# Predicting Infectiousness for Proactive Contact Tracing

Yoshua Bengio[*af]   Prateek Gupta[*abh]   Tegan Maharaj[*ag]   Nasim Rahaman[*ac]   Martin Weiss[*ag]
Tristan Deleu[†af]   Eilif Muller[†af]   Meng Qu[†ai]   Victor Schmidt[†a]   Pierre-Luc St-Charles[†a]
Hannah Alsdurf[‡d]   Olexa Bilanuik[‡a]   David Buckeridge[‡e]   Gáetan Marceau Caron[‡a]
Pierre-Luc Carrier[‡a]   Joumana Ghosn[‡a]   Satya Ortiz-Gagne[‡a]   Chris Pal[‡a]   Irina Rish[‡af]
Bernhard Schölkopf[‡c]   Abhinav Sharma[‡e]   Jian Tang[‡ai]   Andrew Williams[‡a]

## Abstract

The COVID-19 pandemic has spread rapidly worldwide, overwhelming manual contact tracing in many countries and resulting in widespread lockdowns for emergency containment. Large-scale **digital contact tracing (DCT)**[1] has emerged as a potential solution to resume economic and social activity while minimizing spread of the virus. Various DCT methods have been proposed, each making trade-offs between privacy, mobility restrictions, and public health. The most common approach, **binary contact tracing (BCT)**, models infection as a binary event, informed only by an individual's test results, with corresponding binary recommendations that either all or none of the individual's contacts quarantine. BCT ignores the inherent uncertainty in contacts and the infection process, which could be used to tailor messaging to high-risk individuals, and prompt proactive testing or earlier warnings. It also does not make use of observations such as symptoms or pre-existing medical conditions, which could be used to make more accurate infectiousness predictions. In this paper, we use a recently-proposed COVID-19 epidemiological simulator to develop and test methods that can be deployed to a smartphone to locally and proactively predict an individual's infectiousness (risk of infecting others) based on their contact history and other information, while respecting strong privacy constraints. Predictions are used to provide personalized recommendations to the individual via an app, as well as to send anonymized messages to the individual's contacts, who use this information to better predict their own infectiousness, an approach we call **proactive contact tracing (PCT)**. Similarly to other works, we find that compared to no tracing, all DCT methods tested are able to reduce spread of the disease and thus save lives, even at low adoption rates, strongly supporting a role for DCT methods in managing the pandemic. Further, we find a deep-learning based PCT method which improves over BCT for equivalent average mobility, suggesting PCT could help in safe re-opening and second-wave prevention.

## 1 Introduction

Until pharmaceutical interventions such as a vaccine become available, control of the COVID-19 pandemic relies on nonpharmaceutical interventions such as lockdown and social distancing. While these have often been successful in limiting spread of the disease in the short term, these restrictive measures have important negative social, mental health, and economic impacts. **Digital contact tracing (DCT)**, a technique to track the spread of the virus among individuals in a population using smartphones, is an attractive potential solution to help reduce growth in the number of cases and thereby allow more economic and social activities to resume while keeping the number of cases low.

Most currently deployed DCT solutions use **binary contact tracing (BCT)**, which sends a quarantine recommendation to all recent contacts of a person after a positive test result. While BCT is simple

---

[1] All **bolded** terms are defined in the Glossary; Appendix 1.

*,†,‡ Equal contributions, alphabetically sorted; [a]Mila, Québec; [b]University of Oxford; [c]Max-Planck Institute for Intelligent Systems Tübingen; [d]University of Ottawa; [e] McGill University; [f]Université de Montreal; [g]École Polytechnique de Montreal; [h]The Alan Turing Institute; [i]HEC Montréal

and fast to deploy, and most importantly can help curb spread of the disease (Abueg et al., 2020), epidemiological simulations by Hinch et al. (2020) suggest that using only one bit of information about the infection status can lead to quarantining many healthy individuals while failing to quarantine infectious individuals. Relying only on positive test results as a trigger is also inefficient for a number of reasons: (i) Tests have high false negative rates (Li et al., 2020); (ii) Tests are administered late, only after symptoms appear, leaving the asymptomatic population, estimated 20%-30% of cases (Gandhi et al., 2020), likely untested; (iii) It is estimated that infectiousness is highest *before* symptoms appear, well before someone would get a test (Heneghan et al., 2020), thus allowing them to infect others before being traced, (iv) Results typically take at least 1-2 days, and (v) In many places, tests are in limited supply.

Recognizing the issues with test-based tracing, Gupta et al. (2020) propose a rule-based system leveraging other input clues potentially available on a smartphone (e.g. symptoms, pre-existing medical conditions), an approach they call **feature-based contact tracing (FCT)**. Probabilistic (non-binary) approaches, using variants of belief propagation in graphical models, e.g. (Baker et al., 2020; Satorras & Welling, 2020; Briers et al., 2020), could also make use of features other than test results to improve over BCT, although these approaches rely on knowing the social graph, either centrally or via distributed exchanges between nodes. The latter solution may require many bits exchanged between nodes (for precise probability distributions), which is challenging both in terms of privacy and bandwidth. Building on these works, we propose a novel FCT methodology we call **proactive contact tracing (PCT)**, in which we use the type of features proposed by Gupta et al. (2020) as inputs to a predictor trained to output *proactive* (before current-day) estimates of expected infectiousness (i.e. risk of having infected others in the past and of infecting them in the future). The challenges of privacy and bandwidth motivated our particular form of **distributed inference** where we pretrain the predictor offline and do not assume that the messages exchanged are probability distributions, but instead just informative inputs to the node-level predictor of infectiousness.

We use a recently proposed COVID-19 agent-based simulation testbed (Gupta et al., 2020) called COVI-AgentSim to compare PCT to other contact tracing methods under a wide variety of conditions. We develop deep learning predictors for PCT in concert with a professional app-development company, ensuring inference models are appropriate for legacy smartphones. By leveraging the rich individual-level data produced by COVI-AgentSim to train predictors offline, we are able to perform individual-level infectiousness predictions locally to the smartphone, with sensitive personal data never required to leave the device. We find deep learning based methods to be consistently able to reduce the spread of the disease more effectively, at lower cost to mobility, and at lower adoption rates than other predictors. These results suggest that deep learning enabled PCT could be deployed in a smartphone app to help produce a better trade-off between the spread of the virus and the economic cost of mobility constraints than other DCT methods, while enforcing strong privacy constraints.

## 1.1 SUMMARY OF TECHNICAL CONTRIBUTIONS

1. We examine the consequential problem of COVID-19 infectiousness prediction and propose a new method for contact tracing, called proactive contact tracing (see Sec. 2).

2. In order to perform distributed inference with deep learning models, we develop an architectural scaffold whose core is any set-based neural network. We embed two recently proposed networks, namely Deep Sets (Zaheer et al., 2017) and Set Transformers (Lee et al., 2018) and evaluate the resulting models via the COVI-AgentSim testbed (Gupta et al., 2020) (see Sec. 3.1).

3. To our knowledge the combination of techniques in this pipeline is entirely novel, and of potential interest in other settings where privacy, safety, and domain shift are of concern. Our training pipeline consists of training an ML infectiousness predictor on the domain-randomized output of an agent-based epidemiological model, in several loops of retraining to mitigate issues with (i) non-stationarity and (ii) distributional shift due to predictions made by one phone influencing the input for the predictions of other phones. (see Sec. 3.2)

4. To our knowledge this is the first work to apply and benchmark a deep learning approach for probabilistic contact tracing and infectiousness risk assessment. We find such models are able to leverage weak signals and patterns in noisy, heterogeneous data to better estimate infectiousness compared to binary contact tracing and rule-based methods (see Sec. 4)

## 2 PROACTIVE CONTACT TRACING

**Proactive contact tracing (PCT)** is an approach to digital contact tracing which leverages the rich suite of features potentially available on a smartphone (including information about symptoms, preexisting conditions, age and lifestyle habits if willingly reported) to compute proactive estimates of an individual's expected infectiousness. These estimates are used to (a) provide an individualized recommendation and (b) propagate a graded risk message to other people who have been in contact with that individual (see Fig. 1). This stands in contrast with existing approaches for contact tracing, which are either binary (recommending all-or-nothing quarantine to contacts), or require centralized storage of the contact graph or other transfers of information which are incompatible with privacy constraints in many societies. Further, the estimator runs locally on the individual's device, such that any sensitive information volunteered does not need to leave the device.

In Section 2.1, we formally define the general problem PCT solves. In Section 2.2, we describe how privacy considerations inform and shape the design of the proposed framework and implementation. Finally, in Section 3.1, we introduce deep-learning based estimators of expected infectiousness, which we show in Section 4 to outperform DCT baselines by a large margin.

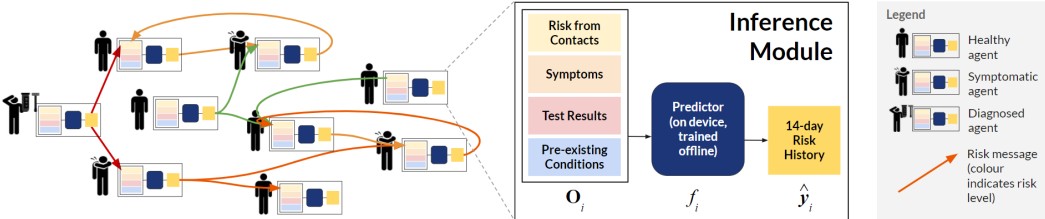

Figure 1: **Proactive contact tracing overview.** Diagram showing **Left:** the propagation of anonymized, graded (non-binary) risk messages between users and **Inset:** overview of the inference module deployed to each user's phone. The inference module for agent $i$ takes in observables $\mathbf{O}_i$, and uses a pretrained predictor $f$ to estimate that agent's risk (expected infectiousness) for each of the last 14 days. Anonymized selected elements of this risk vector are sent as messages to appropriate contacts, allowing them to proactively update their own estimate of expected infectiousness.

### 2.1 PROBLEM SETUP

We wish to estimate **infectiousness** $y_i^{d'}$ of an agent $i$ on day $d'$, given access to *locally observable information* $\mathbf{O}_i^d$ now on day $d \geq d'$ and over the past $d_{max}$ days ($d' \geq d - d_{max}$) Some of the information available on day $d$ is static, including reported age, sex, pre-existing conditions, and lifestyle habits (e.g. smoking), denoted $g_i$, the health profile. Other information is measured each day: the health status $h_i^d$ of reported symptoms and known test results. Finally, $\mathbf{O}_i^d$ also includes information about encounters in the last $d_{max}$ days, grouped in $e_i^d$ for day $d$. Thus:

$$\mathbf{O}_i^d = (g_i, h_i^d, e_i^d, h_i^{d-1}, e_i^{d-1}, \ldots h_i^{d-d_{max}}, e_i^{d-d_{max}}) \tag{1}$$

The information $e_i^{d'}$ about the encounters from day $d'$ is subject to privacy constraints detailed in Section 2.2 but provides indications about the estimated infectiousness of the persons encountered at $d'$, given the last available information by these contacts as of day $d$, hence these contacts try to estimate their past infectiousness a posteriori. Our goal is thus to model the *history* of the agent's infectiousness (in the last $d_{max}$ days), which is what enables the recommendations of PCT to be *proactive* and makes it possible for an infected asymptomatic person to receive a warning from their contact even before they develop symptoms, because their contact obtained sufficient evidence that they were contagious on the day of their encounter. Formally, we wish to model $P_\theta(\mathbf{y}_i^d | \mathbf{O}_i^d)$, where $\mathbf{y}_i^d = (y_i^d, y_i^{d-1}, \ldots, y_i^{d-d_{max}})$ is the vector of present and past infectiousness of agent $i$ and $\theta$ specifies the parameters of the predictive model. In our experiments we only estimate conditional expectations with a predictor $f_\theta$, with $\hat{\mathbf{y}}_i^d = (\hat{y}_i^d, \ldots, \hat{y}_i^{d-d_{max}}) = f_\theta(\mathbf{O}_i^d)$ an estimate of the conditional expected per-day present and past infectiousness $\mathbb{E}_{P_\theta}[\mathbf{y}_i^d | \mathbf{O}_i^d]$.

The predicted expected values are used in two ways. First, they are used to generate messages transmitted on day $d$ to contacts involved in encounters on day $d' \in (d - d_{max}, d)$. These messages

contain the estimates $\hat{y}_i^{d'}$ of the expected infectiousness of $i$ at day $d'$, quantized to 4 bits for privacy reasons discussed in section 2.2. Second, the prediction for today $\hat{y}_i^d$ is also used to form a discrete recommendation level $\zeta_i^d \in \{0, 1, ..., n\}$ regarding the behavior of agent $i$ at day $d$ via a recommendation mapping $\psi$, i.e. $\zeta_i^d = \psi(\hat{y}_i^d)$. [2] At $\zeta_i^d = 0$ agent $i$ is not subjected to any restrictions, $\zeta_i^d = 1$ is baseline restrictions of a post-lockdown scenario (as in summer 2020 in many countries), $\zeta_i^d = n$ is full quarantine (also the behaviour recommended for contacts of positively diagnosed agents under BCT), and intermediate levels interpolate between levels 1 and $n$.

Here we make two important observations about contact tracing: (1) There is a trade-off between decelerating spread of disease, measured by the **reproduction number** $R$, or as number of cases,[3] and minimizing the degree of restriction on agents, e.g., measured by the average number of contacts between agents. Managing this tradeoff is a social choice which involves not just epidemiology but also economics, politics, and the particular weight different people and nations may put on individual freedoms, economic productivity and public health. The purpose of PCT is to improve the corresponding **Pareto frontier**. A solution which performs well on this problem will encode a policy that contains the infection while applying minimal restrictions on healthy individuals, but it may be that different methods are more appropriate depending on where society stands on that tradeoff. (2) A significant challenge comes from the feedback loop between agents: observables $\mathbf{O}_i^d$ of agent $i$ depend on the predicted infectiousness histories and the pattern of contacts generated by the behavior $\zeta_j$ of *other agents* $j$. This is compounded by privacy restrictions that prevent us from knowing which agent sent which message; we discuss our proposed solution in the following section.

## 2.2 PRIVACY-PRESERVING PCT

One of the primary concerns with the transmission and centralization of contact information is that someone with malicious intent could identify and track people. Even small amounts of personal data could allow someone to infer the identities of individuals by cross-referencing with other sources, a process called **re-identification** (El Emam et al., 2011). Minimizing the number of bits being transmitted and avoiding information that makes it easy to triangulate people is a protection against **big brother attacks**, where a central authority with access to the data can abuse its power, as well as against **little brother attacks**, where malicious individuals (e.g. someone you encounter) could use your information against you. Little brother attacks include **vigilante attacks**: harassment, violence, hate crimes, or stigmatization against individuals which could occur if infection status was revealed to others. Sadly, there have been a number of such attacks related to COVID-19 (Russell, 2020). To address this, PCT operates with (1) no central storage of the contact graph; (2) de-identification (Sweeney, 2002) and encryption of all data leaving the phones; (3) informed and optional consent to share information (only for the purpose of improving the predictor); and (4) distributed inference which can achieve accurate predictions without the need for a central authority. Current solutions (Woodhams, 2020) share many of these goals, but are restricted to only binary CT. Our solution achieves these goals while providing individual-level graded recommendations.

Each app records **contacts** which are defined by Health Canada (2020) as being an encounter which lasts at least 15 minutes with a proximity under 2 meters. Once every 6 hours, each application processes contacts, predicts infectiousness for days $d$ to $d_{max}$, and may update its current recommended behavior. If the newly predicted infectiousness history differs from the previous prediction on some day (e.g. typically because the agent enters new symptoms, receives a negative or positive test result, or receives a significantly updated infectiousness estimate), then the app creates and sends small update messages to all relevant contacts. These heavily quantized messages reduce the network bandwidth required for message passing as well as provide additional privacy to the individual. We follow the communication and security protocol for these messages introduced by Alsdurf et al. (2020). We note that a naive method would tend to *over-estimate* infectiousness in these conditions, because repeated encounters with the same person (a very common situation, for example people living in the same household) should carry a *lower* predicted infectiousness than the same number of encounters with different people. To mitigate this over-estimation while not identifying anyone, messages are clustered based on time of receipt and risk level as in Gupta et al. (2020).

---

[2]The recommendations for each level are those proposed by Gupta et al. (2020), hand-tuned by behavioural experts to lead to a reduction in the number of contacts; details in Appendix 5. Full compliance with recommendations is not assumed.

[3]Note that the the number of cases is highly non-stationary; it grows exponentially over time, even where $R$, which is in the exponent, is constant.

## 3 METHODOLOGY FOR INFECTIOUSNESS ESTIMATION

### 3.1 DISTRIBUTED INFERENCE

Distributed inference networks seek to estimate marginals by passing messages between nodes of the graph to make predictions which are consistent with each other (and with the local evidence available at each node). Because messages in our framework are highly constrained by privacy concerns in that we are prevented from passing distributions or continuous values between identifiable nodes, typical distributed inference approaches (e.g. loopy belief propagation (Murphy et al., 1999), variational message passing (Winn & Bishop, 2005), or expectation propagation (Minka, 2001)) are not readily applicable. We instead propose an approach to distributed inference which uses a trained ML predictor for estimating these marginals (or the corresponding expectations). The predictor $f$ is trained offline on simulated data to predict expected infectiousness from the locally available information $\mathbf{O}_i^d$ for agent $i$ on day $d$. The choice of ML predictor is informed by several factors. First and foremost, we expect the input $e_i^d$ (a component of $\mathbf{O}_i^d$) to be variable-sized, because we do not know *a priori* how many contacts an individual will have on any day. Further, privacy constraints dictate that the contacts within the day cannot be temporally ordered, implying that the quantity $e_i^d$ is set-valued. Second, given that we are training the predictor on a large amount of domain-randomized data from many epidemiological scenarios (see Section 3.2), we require the architecture to be sufficiently expressive. Finally, we require the predictor to be easily deployable on edge devices like legacy smartphones.

To these ends, we construct a general architectural scaffold in which any neural network that maps between sets may be used. In this work, we experiment with Set Transformers (Lee et al., 2018) and a variant of Deep Sets (Zaheer et al., 2017). The former is a variant of Transformers (Vaswani et al., 2017), which model pairwise interactions between all set elements via multi-head dot-product attention and can therefore be quite expressive, but scales quadratically with elements. The latter relies on iterative max-pooling of features along the elements of the set followed by a broadcasting of the aggregated features, and scales linearly with the number of set elements (see Figure 2).

In both cases, we first compute two categories of embeddings: per-day and per-encounter. Per-day, we have embedding MLP modules $\phi_h$ of health status, $\phi_g$ for the health profile, and a linear map $\phi_{\delta d}$ for the day-offset $\delta d = d - d'$. Per-encounter, the cluster matrix $e_i^d$ can be expressed as the set $\{(r_{j \to i}^d, n_{j \to i}^d)\}_{j \in K_i^d}$ where $j$ is in the set $K_i^d$ of putative anonymous persons encountered by $i$ at $d$, $r_{j \to i}$ is the risk level sent from $j$ to $i$ and $n_{j \to i}$ is the number of repeated encounters between $i$ and $j$ at $d$. Accordingly, we define the risk level embedding function $\phi_e^{(r)}$ and $\phi_e^{(n)}$ for the number of repeated encounters. $\phi_e^{(r)}$ is parameterized by an embedding matrix, while $\phi_e^{(n)}$ by the vector

$$(\phi_e^{(n)}(n))_{2i} = \sin\left(n/10000^i\right) \text{ and } (\phi_e^{(n)}(n))_{2i+1} = \cos\left(n/10000^i\right) \tag{2}$$

which resembles the positional encoding in (Vaswani et al., 2017) and counteracts the spectral bias of downstream MLPs (Rahaman et al., 2019; Mildenhall et al., 2020). We tried alternative encoding schemes like thermometer encodings (Buckman et al., 2018), but they did not perform better in our experiments. Next, we join the per-day and per-encounter embeddings to obtain $\mathcal{D}_i^d$ and $\mathcal{E}_i^d$ respectively. Where $\otimes$ is the concatenation operation, we have:

$$\mathcal{D}_i^d = \{\phi_h(h_i^{d'}) \otimes \phi_g(g_i) \otimes \phi_{\delta d}(d' - d) \,|\, d' \in \{d, ..., d - d_{max}\}\} \tag{3}$$

$$\mathcal{E}_i^d = \{\phi_e^{(r)}(r_{j \to i}^{d'}) \otimes \phi_e^{(n)}(n_{j \to i}^{d'}) \otimes \phi_h(h_i^{d'}) \otimes \phi_{\delta d}(d' - d) \,|\, j \in K_i^{d'}, \forall d' \in \{d, ..., d - d_{max}\}\} \tag{4}$$

The union of $\mathcal{D}_i^d$ and $\mathcal{E}_i^d$ forms the input to a set neural network $f_S$, that predicts infectiousness:

$$\hat{\mathbf{y}}_i^d = f_S(\mathcal{O}_i^d) \text{ where } \mathcal{O}_i^d = \mathcal{D}_i^d \cup \mathcal{E}_i^d \text{ and } \hat{\mathbf{y}}_i^d \in \mathbb{R}_+^{d_{max}} \tag{4}$$

The trunk of both models – deep-set (DS-PCT) and set-transformer (ST-PCT) – is a sequence of 5 set processing blocks (see Figure 2). A subset of outputs from these blocks (corresponding to $\mathcal{D}_i^d$) is processed by a final MLP to yield $\hat{\mathbf{y}}_i^d$. As a training objective for agent $i$, we minimize the Mean Squared Error (MSE) between $\mathbf{y}_i^d$ (which is generated by the simulator) and the prediction $\hat{\mathbf{y}}_i^d$. We treat each agent $i$ as a sample in batch to obtain the sample loss $L_i = \text{MSE}(\mathbf{y}_i^d, \hat{\mathbf{y}}_i^d) = \frac{1}{d_{max}} \sum_{d'=d-d_{max}}^{d} (y_i^{d'} - \hat{y}_i^{d'})^2$. The net loss is the sum of $L_i$ over all agents $i$.

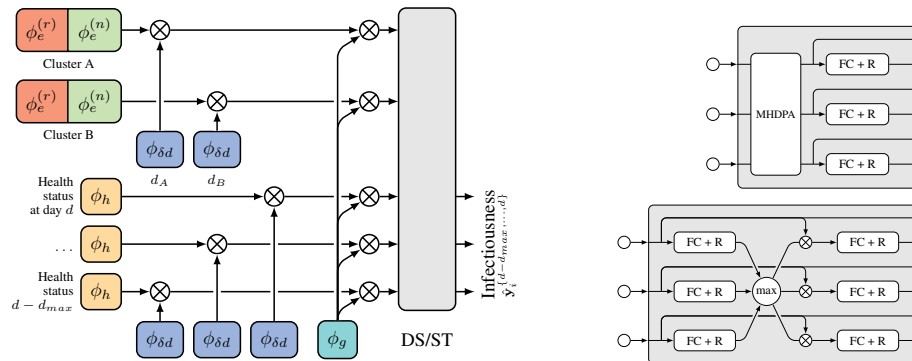

Figure 2: **PCT model architecture.** Diagram showing **Left**: The embedding network combining pre-existing conditions ⬤, day offsets ⬤, risk message clusters ⬤ ⬤, and symptoms information ⬤ to be fed into a stack of 5 either Deep-Set (DS) or Set Transformer (ST) blocks ⬤. **Right-top**: (ST) self-attention block featuring Multi-Head Dot Product Attention (MHDPA), Fully-Connected layers with ReLU (FC + R) and residual connections. **Right-bottom:** (DS) set processing block. Here, the $\otimes$ denotes concatenation and the $\oplus$ addition operation.

## 3.2 TRAINING PROCEDURE

Unlike existing contact tracing methods like BCT and the NHSx contact tracing app (Briers et al., 2020) that rely on simple and hand-designed heuristics to estimate the risk of infection, our core hypothesis is that methods using a machine-learning based predictor can learn from patterns in rich but noisy signals that might be available locally on a smartphone. In order to test this hypothesis prior to deployment in the real world, we require a simulator built with the objective to serve as a testbed for app-based contact tracing methods. We opt to use COVI-AgentSim (Gupta et al., 2020), which features an agent specific virology model together with realistic agent behaviour, mobility, contact and app-usage patterns. With the simulator, we generate large datasets of $\mathcal{O}(10^7)$ samples comprising the input and target variables defined in sections 2.1 and 3.1. Most importantly for our prediction task, this includes a continuous-valued infectiousness parameter as target for each agent.

We use $240$ runs of the simulator to generate this dataset, where each simulation is configured with parameters randomly sampled from selected intervals (See Appendix 5.2). These parameters include the adoption rate of the contact tracing app, initial fraction of exposed agents in the population, the likelihood of agents ignoring the recommendations, and strength of social distancing and mobility restriction measures. There are two reasons for sampling over parameter ranges: First, the intervals these parameters are sampled from reflect both the intrinsic uncertainty in the parameters, as well as the practical setting and limitations of an app-based contact tracing method (e.g. we only care about realistic levels of app usage). Second, randomly sampling these key parameters significantly improves the diversity of the dataset, which in turn yields predictors that can be more robust when deployed in the real world. The overall technique resembles that of **domain-randomization** (Tobin et al., 2017; Sadeghi & Levine, 2016) in the Sim2Real literature, where its efficacy is well studied for transfer from RGB images to robotic control, e.g. (Chebotar et al., 2018; OpenAI et al., 2018).

We use 200 runs for training and the remaining 40 for validation (full training and other reproducibility details are in Appendix 5). The model with the best validation score is selected for *online* evaluation, wherein we embed it in the simulator to measure the reduction in the $R$ as a function of the average number of contacts per agent per day. While the evaluation protocol is described in section 4, we now discuss how we mitigate a fundamental challenge that we share with offline reinforcement learning methods, that of **auto-induced distribution shift** when the model is used in the simulation loop (Levine et al., 2020; Krueger et al., 2020).

To understand the problem, consider the case where the model is trained on simulation runs where PCT is driven by ground-truth infectiousness values, i.e. an *oracle* predictor. The predictions made by an oracle will in general differ from ones made by a trained model, as will resulting dynamics of spread of disease. This leads to a distribution over contacts and epidemiological scenarios that

the model has not encountered during training. For example, oracle-driven PCT might even be successful in eliminating the disease in its early phases – a scenario unlikely to occur in model-driven simulations. For similar reasons, oracle-driven simulations will also be less diverse than model-driven ones. To mitigate these issues, we adopt the following strategy: First, we generate an initial dataset with simulations driven by a *noisy-oracle*, i.e. we add multiplicative and additive noise to the ground-truth infectiousness to partially emulate the output distribution resulting from trained models. The corresponding noise levels are subject to domain-randomization (as described above), resulting in a dataset with some diversity in epidemiological scenarios and contact patterns. Having trained the predictor on this dataset (until early-stopped), we generate another dataset from simulations driven by the thus-far trained predictor, in place of the noisy-oracle. We then fine-tune the predictor on the new dataset (until early stopped) to obtain the final predictor. This process can be repeated multiple times, in what we call **iterative retraining**. We find three steps yields a good trade-off between performance and compute requirement.

## 4    EXPERIMENTS

We evaluate the proposed PCT methods, **ST-PCT** and **DS-PCT**, and benchmark them against: test-based BCT; a rule-based FCT **Heuristic** method proposed by Gupta et al. (2020)[4]; and a baseline **No Tracing (NT)** scenario which corresponds to recommendation level 1 (some social distancing).

*EXP1*: In Figure 3, we plot the Pareto frontier between spread of disease ($R$) and the amount of restriction imposed. To traverse the frontier at 60% adoption rate, we sweep through multiple values of the simulator's **global mobility scaling factor**, a parameter which controls the strength of social distancing and mobility restriction measures. Each method uses 3000 agents for 50 days with 12 random seeds, and we plot the resulting number of contacts per day per human. We fit a Gaussian Process Regressor to the simulation outcomes and highlight the average reduction in $R$ obtained in the vicinity of $R \approx 1$, finding DS-PCT and ST-PCT reduce $R$ to below 1 at a much lower cost to mobility on average. For subsequent plots, we select the number of contacts per day per human such that the no-tracing baseline yields a realistic post-lockdown $R$ of around 1.2 (Brisson & et al., 2020)[5].

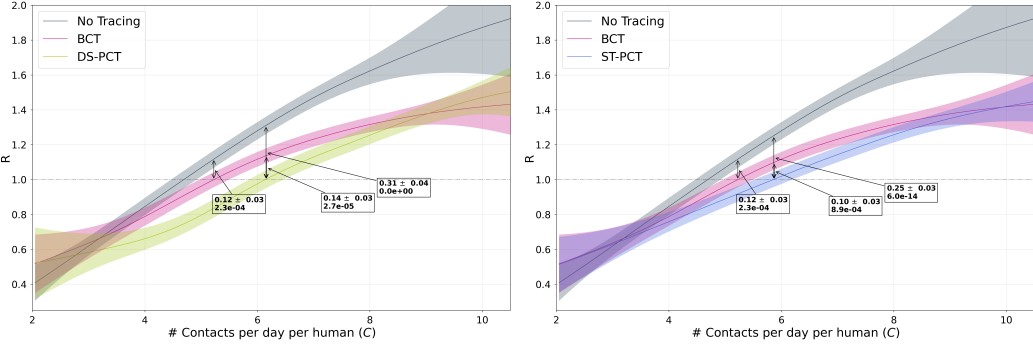

Figure 3: **Pareto frontier of mobility and disease spread:** reproduction number $R$ as a function of mobility. In boxes, average difference +/- standard error, p-value under null hypothesis of no difference. Note that small differences in $R$ over time produce large changes to the number of cases. **Gist:** All methods span a wide range of $R$ values in their Pareto frontier, including values for the No Tracing scenario which have $R$ below 1, achieved by imposing strong restrictions on mobility (e.g. a lockdown). However DCT methods are able to reduce $R$ at much lower cost to average mobility. Compared to NT, BCT has a 12% advantage in $R$, DS-PCT (**left**) 31% and ST-PCT 25% (**right**).

*EXP2*: Figure 4 compares various DCT methods in terms of their cumulative cases and their fraction of **false quarantine**: the number of healthy agents the method wrongly recommends to quarantine.

---

[4]While we did experiment with additional methods like linear regression and MLPs, they did not improve the performance over the rule-based FCT heuristic proposed by Gupta et al. (2020).

[5]To estimate this number, we use the GP regression fit in figure 3 and consider the $x$ value for which the mean of the NT process is at 1.2. We find an estimate at 5.61, and select only the runs yielding effective number of contacts that lie within the interval $(5.61 - 0.5, 5.61 + 0.5)$.

Again, all DCT methods have a clear advantage over the no-tracing baseline, while the number of agents wrongly recommended quarantine is much lower for ML-enabled PCT. [6]

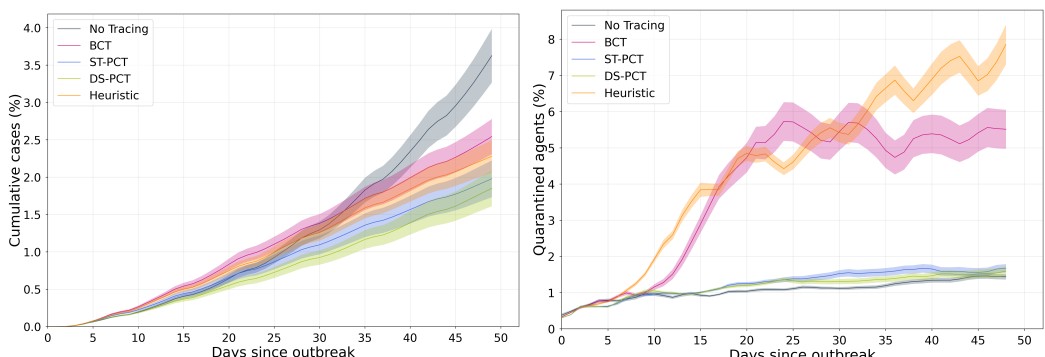

Figure 4: **Left:** Cumulative case counts for each method, 60% adoption, 50 days, with all runs normalized to $5.61 \pm 0.5$ effective contacts per day per agent. **Right:** Mobility restriction for the same experiments (fraction of quarantines). **Gist:** For a similar number of cumulative cases to other DCT methods (left), PCT methods impose very little mobility restriction (right), close to NT.

***EXP3****:* In Figure 5 **Left** we plot the bootstrapped distribution of mean $R$ for different DCT methods. Recall $R$ is an estimate of how many other agents an infectious agent will infects, i.e. even a numerically small improvement in $R$ could yield an exponential improvement in the number of cases. We find that both PCT methods yield a clear improvement over BCT and the rule-based heuristic, all of which significantly improve over the no-tracing baseline. We hypothesize this is because deep neural networks are better able to capture the non-linear relationship between features available on the phone, interaction patterns between agents, and individual infectiousness.

***EXP4****:* In Figure 5 **Right** we investigate the effect of **iterative retraining** on the machine-learning based methods, DS-PCT and ST-PCT. We evaluate all 3 iterations of each method in the same experimental setup as in Figure 5 and find that DS-PCT benefits from the 3 iterations, while ST-PCT saturates at the second iteration. We hypothesize that this is a form of overfitting, given that the set transformer (ST) models all-to-all interactions and is therefore more expressive than deep-sets (DS).

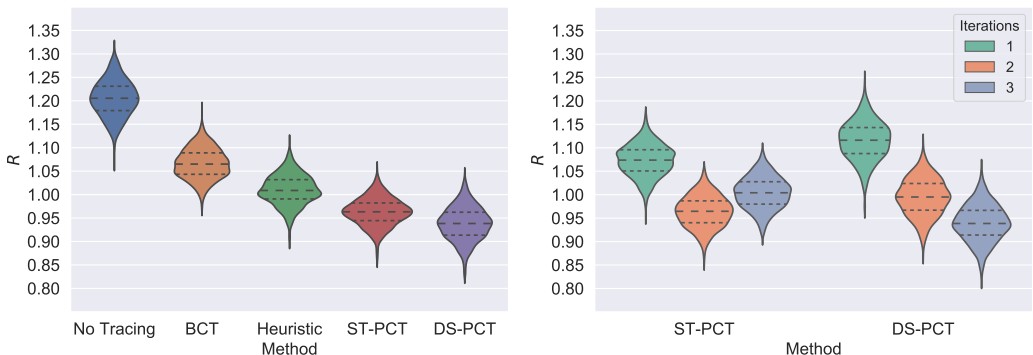

Figure 5: **Method Comparison and Retraining**. **Left:** Distribution (median and quartiles) of $R$ at 60% adoption where the ML methods outperform the Heuristic and BCT methods proposed by Gupta et al. (2020). **Right:** Iterative re-training significantly improves model performance. Upon the second re-training, however, it seems that ST-PCT begins to overfit.

---

[6]Note that the baseline no tracing method also has false quarantines, because household members of an infected individual are also recommended quarantine, irrespective of whether they are infected.

**_EXP5_**: In Figure 6, we analyze the sensitivity of the various methods to adoption rate, which measures what percent of the population actively uses the CT app. Adoption rate is an important parameter for DCT methods, as it directly determines the effectiveness of an app. We visualize the effect of varying the adoption rate on the reproduction number $R$. Similarly to prior work (Abueg et al., 2020), we find that all DCT methods improve over the no-tracing baseline even at low adoptions and PCT methods dominate at all levels of adoption.

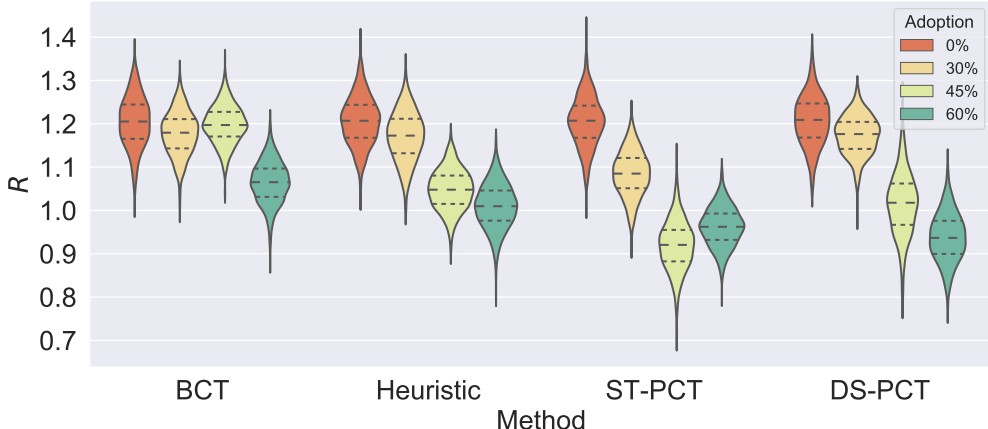

Figure 6: **Adoption rate comparison**. We compare all methods for adoption rates between 0% (NT) and 60%. **Gist:** All methods are able to improve over NT, even at low adoption rates. At 30% and 45%, ST-PCT performs the best by a relatively wide margin while DS-PCT outperforms it at 60%.

## 5 CONCLUSION

Our results demonstrate the potential benefit of digital contact tracing approaches for saving lives while reducing mobility restrictions and preserving privacy, independently confirming previous reports (Ferretti et al., 2020; Abueg et al., 2020). Of all methods in our study, we find that deep learning based PCT provides the best trade-off between restrictions on mobility and reducing the spread of disease under a range of settings, making it a potentially powerful tool for saving lives in a safe deconfinement. This area of research holds many interesting avenues of future work in machine learning, including: (1) The comparison of methods for fitting parameters of the epidemiological simulator data using spatial information (not done here to avoid the need for GPS), (2) Using reinforcement learning to learn (rather than expert hand-tune) the mapping from estimated infectiousness and individual-level features (number of contacts, age, etc.), and thereby target desirable outcomes for heterogeneous populations.

Accurate and practical models for contact tracing are only a small part of the non-pharmaceutical efforts against the pandemic, a response to which is a complex venture necessitating cooperation between public health, government, individual citizens, and scientists of many kinds - epidemiologists, sociologists, behavioural psychologists, virologists, and machine learning researchers, among many others. We hope this work can play a role in fostering this necessary collaboration.

ACKNOWLEDGEMENTS

We thank all members of the broader COVI project `https://covicanada.org` for their dedication and teamwork; it was a pleasure and honour to work with such a great team. We also thank the Mila and Empirical Inference communities, in particular Xavier Bouthillier, Vincent Mai, Gabriele Prato, and Georgios Arvanitidis, for detailed and helpful feedback on early drafts. The authors gratefully acknowledge the following funding sources: NSERC, IVADO, CIFAR. This project could not have been completed without the resources of MPI-IS cluster, Compute Canada & Calcul Quebec, in particular the Beluga cluster. In gratitude we have donated to a project to help understand and protect the St. Lawrence beluga whales for whom the cluster is named `https://baleinesendirect.org/`

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

APPENDIX 1: GLOSSARY OF BOLDED TERMS

**Contact tracing**    Finding the people who have been in contact with an infected person, typically using information such as test results and phone surveys, and recommending that they quarantine themselves to prevent further spread of the disease.

**Manual contact tracing**    Method for contact tracing using trained professionals who interview diagnosed individuals to identify people that have come into contact with an infected person and recommend a change in their behavior (or further testing).

**Digital contact tracing (DCT)**    Predominantly smartphone-based methods of identifying individuals at high risk of contracting an infectious disease based on their interactions, mobility patterns, and available medical information such as test results.

**Binary contact tracing (BCT)**    Methods which use binary information (e.g. tested positive or negative) to perform contact tracing, thus putting users in binary risk categories (at-risk or not-at-risk).

**Proactive contact tracing (PCT)**    DCT methods which produce graded risk information to mitigate spread of disease, thus putting users in graded risk categories (more-at-risk or less-at-risk), potentially using non-binary clues like reported symptoms which are available earlier than test results.

**Agent-based epidemiological models (ABMs)**    Also called individual-based, this type of simulation defines rules of behaviour for individual agents, and by stepping forward in time, contact patterns between agents are generated in a "bottom up" fashion. Contrast with **Population-level epidemiological models**.

**Risk of infection**    Expected infectiousness, or probability of infection for a susceptible agent by an infectious agent given a qualifying contact (15+ minutes at under 2 meters).

**Contact**    An encounter between 2 agents which lasts at least 15 minutes with a distance under 2 meters (Health Canada, 2020).

**Risk Level**    A version of the risk of infection quantized into 16 bins (for reducing the number of bits being exchanged, for better privacy protection). The thresholds are selected by running the domain randomization with separate seeds and grouping risk messages such that there are an approximately uniform number in each bin.

**Risk mapping**    A table which maps floating point risk scalars into one of 16 discrete risk levels.

**Recommendation mapping**    A table of which recommendation level should be associated with a given risk level. Each recommendation level comes with a series of specific recommendations (e.g. "Limit contact with others", "wash hands frequently", "wear a mask when near others", "avoid public transportation", etc) which should be shown for a given risk level. Reminders of these recommendations may be sent via push notification more or less often depending on the recommendation level.

**Population-level epidemiological models**    This type of model fits global statistics of a population with a mathematical model, typically sets of ordinary differential equations. The equations track some statistics over time, typically counts of people in each of a few "compartments", e.g. the number of Infected and Recovered, and the parameters of the equations are often tuned to match statistics collected from real data. It does not give any information about the individual-level contact patterns which give rise to these statistics.

**Compartment model (SEIR)**    A compartment model tracks the counts of agents in each of several mutually-exclusive categories called "compartments". In an SEIR model, the 4 compartments are **Susceptible, Exposed, Infectious, and Recovered** (see entries for each of these words).

**Susceptible**    At risk of catching disease, but not infected.

**Exposed or infected**    Infected with the disease, (i.e. potentially carries some **viral load**).

**Infectious**    Carries sufficient viral load to transmit the disease to others. In real life, this typically means that the virus has multiplied sufficiently to overwhelm the immune system.

**Recovered**    No longer carries measurable viral load, after having been infected. In real life this is typically measured by two successive negative **lab tests**.

**Viral load**    Viral load is the number of actual viral RNA in a person as measured by a **lab test**.

**Effective viral load**    A term we introduce representing a number between 0 and 1 which we use as a proxy for viral load. It could be converted to an actual viral load via multiplying by the maximum amount of viral RNA detectable by a lab test.

**Infectiousness**    Degree to which an agent is able to transmit viruses to another agent. A factor in determining whether an encounter between a susceptible and an infectious agent results in the susceptible agent being exposed (it is a property of the infector agent; this does not consider environmental factors or the other agent's susceptibility; see **Transmission probability**).

**Attack rate**    The proportion of individuals in a population who are infected over some time period.

**Contamination duration**    How long a location remains infectious after an infected person has visited it and shed viruses there.

**Transmission probability**    The chance that one agent infects another during an encounter. **Infectiousness** of the infected agent, **susceptibility** of the other agent, both their behaviour (e.g. usage of masks) and other environmental factors all play a role in determining this probability.

**PCR Test**    Also, colloquially referred to as a lab test. In this context, this means a Polymerase Chain Reaction (PCR) test used to amplify viral DNA samples, typically obtained via a nasopharangeal swab.

**Asymptomatic**    When an infected person shows no symptoms. For COVID-19, many people remain asymptomatic for the entire course of their disease.

**Incubation days**    The number of days from exposure (infection) until an agent shows symptoms. In real life this is the average amount of time it takes the viral RNA to multiply sufficiently in / burst out of host cells and cause an immune response, which is what produces the symptoms.

**Reproduction number ($R$)**    The average number of other agents an agent infects, measured over a certain window of time. We follow Gupta et al. (2020) and approximate $R$ by computing the infection tree and taking the ratio of the number of infected children divided by the number of parents who are recovered infectors.

**Serial interval**    The average number of days between successive symptom onsets (i.e. between the **incubation** time of the infector and the infectee).

**Generation time**    The number of days between successive infections (i.e. between the exposure of the infector and the infectee). Generation time is difficult to measure directly so one can estimate it from the **serial interval**.

**Domain randomization**    A method of generating a training dataset by sampling from several distributions, each corresponding to a different setting of parameters of a simulator. This allows to generate a dataset which covers a potentially wide range of settings, improving generalization to

environments which may not be well covered by the simulator in any one particular setting chosen a priori.

**Oracle Predictor**     Baseline for the **Set Transformer** which uses ground-truth infectiousness levels as "predictions". Provides an expected upper-bound performance for the predictors and can be used to provide risk level inputs when pre-training a predictors.

**Multi-Layer Perceptron (MLP)**     A risk prediction model which concatenates all inputs (after heuristically aggregating to a fixed size vector the set-valued inputs), feeds these through several fully-connected neural network layers and is trained to predict infectiousness. See Section 5.1.

**Set Transformer (ST) Model**     A risk prediction model which uses a modified set transformer to process and attend to inputs, and is trained to predict infectiousness. See Section 5.1.

**Deep Set (DS) Model**     Similar structure as the Set Transformer, but using pooling instead of self-attention. This allows it to use less memory and compute as compared to the Set Transformer.

**Heuristic (HCT) Model**     A rule-based method for predicting risk histories and corresponding current behavior levels developed in Gupta et al. (2020).

**Adoption rate**     The proportion of the total population which uses a digital contact tracing application.

**Domain randomization**     The technique of varying the parameters of a simulation environment to create a broad distribution of training data.

**Re-identification**     The process of matching anonymous data with publicly available data so as to determine which person is the owner.

**Big brother attacks**     "Big brother" is a reference to the 1984 book by George Orwell where a highly regulated government surveilled and controlled its citizenry. The term "big brother attacks" references the idea that an organized group (e.g., state, federal, academic or financial institution) may try to gain control of your data.

**Little brother attacks**     Little brother attacks references the idea that potential security threats are also posed by smaller actors like individuals, criminals, and data brokers.

**Vigilante attacks**     Extra-judicial violence by an individual. In the context of this paper, specifically as a result of a recommendation shown to the user or gained through other means.

**Degree of Restriction**     Ratio between the number of people given a recommendation in the highest level of restriction (i.e. quarantine) relative to other categories.

**Global mobility scaling factor**     A simulator parameter enabling us to vary the amount of contacts which may lead to infection in the simulator. It allows us to scale the mobility to simulate pre-lockdown or post-lockdown environments.

**Auto-induced distribution shift**     A change in distribution of data observed by an agent or algorithm as a result of the agent or algorithm's actions.

**Iterative re-training**     The process of generating data using some method in a simulator, training a model on this data in a supervised manner, then evaluating the model in the simulator to produce more data.

**Pareto frontier**     The Pareto frontier is a way of evaluating a tradeoffs in a set of policies and environments with multi-dimensional outputs (e.g., viral spread and mobility).

## Appendix 2: Motivating Example

Figure 7: **Motivating example comparing manual, binary, and proactive contact tracing**: This example shows the potential effectiveness of early warnings in controlling the spread of the infection. Manual tracing is delayed because of the time between diagnosis and calling all contacts. Both manual and digital contact tracing are sending late signals because they only make use of the strongest possible signal (positive diagnosis). The proposed ML approach takes advantage of reported symptoms and the propagation of risk signals between phones to obtain much earlier signals.

## Appendix 3: Comparison of approach to related work

**Epidemiological Modeling and Simulations.** Given that modeling contact-tracing requires capturing past interactions, it is mathematically complicated to consider the dynamic nature of network interactions. An agent-based model (simulator), on the other hand, gives us maximum flexibility to incorporate real data and/or assumptions easily and emulate the effect of personalized policies (such as resulting from the proposed app). Models with binary contagion and random-walk mobility are ubiquitous (Stevens, 2020). The appeal of such models is their simplicity; they are easy to code, fast to run, and can give a general picture of some aspects of disease spread. There are other models with increasing complexity of either the mobility model, contagion model, agent demographics, or some combination of these, e.g. (Verity et al., 2020; Vespignani et al., 2020; Wood et al., 2020; Lorch et al., 2020; Hinch et al., 2020).

GLEAM (Global Epidemic And Mobility model) (Vespignani et al., 2020) is an off-the-shelf simulation platform for epidemics which offers mobility patterns and demographic information, and uses a generic format for defining how a disease depends on these two things. Similarly, FRED (Framework for Reconstructing Epidemiological Dynamics) provides an open-source, agent-based model with realistic social networks and US demographics. JJ et al. (2013), and (Wood et al., 2020) build an intervention-planning tool on top of this simulator. These works are comparable to our simulator only; they do not do any contact tracing or individual risk prediction.

The work presented in (Lorch et al., 2020) and (Ferretti et al., 2020) is closely related to ours. A detailed mobility model for interactions is presented in (Lorch et al., 2020), but the epidemiological model of the disease is much simpler there. First, the contact graph is built based on their mobility model. The next step is an implementation of various policy interventions by health authorities, which includes contact tracing. This is done instead of building a contact graph that takes into account policy interventions and contact tracing apps at an individual level. Next, most similarly to our work, (Ferretti et al., 2020) proposes an app-based tracing and recommendations. However, their epidemiological model is a very simple differential-equation model, and interventions for controlling the disease are computed analytically.

**Risk estimation approaches.** While the applications deployed so far are based primarily on binary contact tracing as discussed above, some probabilistic risk estimation approaches, similar to ours, have been developed for other diseases or applications, and have begun to be applied to COVID-19. For example, (Baker et al., 2020) uses the susceptible-infected-recovered (SIR) model and describes the dynamical process of infection propagation using the dynamical message passing equations from (Lokhov et al., 2014); the probability of each node (person) to be in a specific state (S,I or R) is estimated via the dynamic message-passing (DMP) algorithm, which belongs to the

family of local message-passing methods similar to belief propagation (BP) algorithm (Yedidia et al., 2000) for estimating marginal probability distributions over the network nodes; despite BP being only an approximate inference method, not guaranteed to converge to the correct marginals when the underlying graphical model has (undirected) cycles, it demonstrated remarkable performance in various applications. A similar approach based on BP was also discussed in (Murphy, 2020). Furthermore, there are recent extensions of belief propagation approach graph neural nets (Satorras & Welling, 2020). Also, another recent work uses Gibbs sampling and SEIR model for their test-trace-isolate approach (Herbrich et al., 2020). However, such approaches typically rely on the knowledge of the social interaction graph, which is not available in our case due to privacy and security constraints.

To the best of our knowledge, our work is the first to use an approach based on detailed agent-based epidemiological model together with a model of phone app messaging to generate simulated data for training an ML-based predictor of individual-level risk.

**Digital contact tracing for COVID-19** (Hinch et al., 2020; Hellewell et al., 2020; Aleta et al., 2020; Grantz et al., 2020) study, either via simulations or mathematical models, the conditions under which BCT can be effective. Toward addressing the issues with BCT, (Hinch et al., 2020) show in simulation that using self-reported symptoms in addition to test results can greatly help control an outbreak. Probabilistic (non-binary) approaches to the problem of contact tracing (e.g. Baker et al. (2020); Satorras & Welling (2020); Briers et al. (2020)) typically assume full access to location histories and contact graph, an unacceptable violation of privacy in most places in the world. As a result, these methods are most often used for predicting overall patterns of disease spread.

**Agent-Based Models as a generative process** Our use of an agent-based simulator (Gupta et al., 2020) as a generative model allows us to generate fine-grained (continuous) values for expected infectiousness with realistic contact patterns. Most works performing probabilistic inference for disease modeling use a simple differential equation generative model, which does not characterize individual behaviour but rather the dynamics of transition between each of several mutually exclusive disease states. Such models make many simplifying assumptions, such as contact patterns based on random walks, which make them unsuitable for individual-level prediction of infectiousness; they are typically used instead to infer latent variables such as infectiousness that would be consistent with the population-level statistics generated by the differential equation model, and/or to predict statistics of spread of the disease in a population, e.g. (ref, ref). While agent-based models are widely used in epidemiological literature to model the spread of disease (see e.g. (ref) for review), to our knowledge we are the first to use an ABM as a generative model for training a deep learning-based infectiousness predictor.

**Distributed inference and belief propagation** Belief propagation in graphical models is often used for disease spread modeling, e.g. Fan et al. (2016). Some recent works have applied this to COVID-19; for example, Baker et al. (2020) use the susceptible-infected-recovered (SIR) model and describe the process of infection propagation using the dynamical message passing equations from Lokhov et al. (2014). A work concurrent with ours follows a similar justification for modeling a latent parameter of expected infectiousness, using an SEIR model with inference via (Herbrich et al., 2020). However, these approaches rely on a centralized social graph or a large number of bits exchanged between nodes, which is challenging both in terms of privacy and bandwidth. This challenge motivated our particular form of distributed inference where we pretrain the predictor and do not assume that the messages exchanged are probability distributions, but instead just informative input to the node-level predictor.

## Appendix 4: Experimental details

### 5.1 ML architectures and baseline details:

**Binary contact tracing** quarantines app-users who had high-risk encounters with an app-user who receives a positive PCR test. Under BCT1, if Alice gets a positive test result, then every user who encountered Alice within 14 days of her receiving the positive test result is sent a message which places them in app-recommendation level 3 (quarantine) for 14 days. Formally, $\zeta_i^d = \psi(\hat{y}_i^d) = 3$.

**Set Transformer**    Recall that in Section 3 we proposed two parameterizations of the model (DS and ST in Figure 2). In the first proposal, we use a set transformer (ST) to model interactions between the elements in the set $\mathbb{D}_i^d \cup \mathbb{E}_i^d$. We now describe the precise architecture of the model used.

The model comprises 5 embedding modules, namely: the health status embedding $\phi_{hs}$, health profile embedding $\phi_{hp}$, day offset embedding $\phi_{do}$, risk message embedding $\phi_e^{(r)}$ and an embedding $\phi_e^{(n)}$ of the number of repeated encounters.

The model was trained for 160 epochs on a domain randomized dataset (see below) comprising $\sim 10^7$ samples. We used a batch-size of 1024, resulting in $\sim 80k$ training steps. The learning rate schedule is such that the first $2.5k$ steps are used for linear learning-rate warmup, wherein the learning rate is linearly increased from 0 to $2 \times 10^{-4}$, followed by a cosine annealing schedule that decays the learning rate from $2 \times 10^{-4}$ to $8 \times 10^{-6}$ in $50k$ steps.

## 5.2    DOMAIN RANDOMIZATION

Inspired by research in hyper-parameter search (Bergstra & Bengio, 2012) and recent advances in deep reinforcement learning (Tobin et al., 2017) we created the transformer's training data by sampling uniformly in the following ranges:

1.  Adoption rate $\in [30 - 60]$
2.  Carefulness $\in [0.5 - 0.8]$
3.  Initial proportion of exposed people $\in [0.002, 0.006]$
4.  Oracle additive noise $\in [0.05 - 0.15]$
5.  Oracle multiplicative noise $\in [0.2 - 0.8]$
6.  Global mobility scaling factor $\in [0.3 - 0.9]$
7.  Symptoms dropout: Likelihood of not reporting some symptoms $\in [0.1, 0.6]$
8.  Symptoms drop-in: Likelihood of falsely reporting symptoms $\in [0.0001, 0.001]$
9.  Quarantine dropout (test) $\in [0.01, 0.03]$: likelihood of not quarantining when recommended to quarantine due to a positive test
10.  Quarantine dropout (household) $\in [0.02, 0.05]$: likelihood of not quarantining when recommended to quarantine because a household member got a positive test
11.  All-levels dropout $\in [0.01, 0.05]$: likelihood of not following app-recommended behavior and instead exhibit pre-pandemic behavior

## 5.3    TRAINING TIME

Our ML experiments use approximately 250 days training time on GPUs while simulations required approximately 41 days of CPU time. All CPU time was run on compute using renewable resources.

## APPENDIX 5: RELEVANT BACKGROUND ON COVI-AGENTSIM

Details on the simulator can be found in Gupta et al. (2020), and the code is open-source at `https://github.com/mila-iqia/COVI-AgentSim`. Here we summarize some of the most relevant details about the simulator for ease of reference.

### 5.4 APP ADOPTION

COVI-AgentSim models app adoption proportional to smartphone usage statistics, shown in Table 1.

| % of population with app | Uptake required to get that % |
|---|---|
| 1 | 1.50 |
| 30 | 42.15 |
| 40 | 56.18 |
| 60 | 84.15 |
| 70 | 98.31 |

Table 1: **Adoption Rate vs Uptake**: The left column show the total percentage of the population with the app, while the right column shows the proportion of *smartphone users* with the app.

### 5.5 RECOMMENDATION LEVELS AND CONTAINMENT PROTOCOL

In Gupta et al. (2020), the implementation of binary contact tracing uses a simple, binary containment protocol: quarantine at home for 14 days if in contact with someone who has had a positive test result, otherwise do nothing. In contrast, the graded recommendation levels used by feature-based contact tracing methods including Heuristic and Proactive Contact Tracing are not binary, and do not model an explicit containment protocol. Instead, Gupta et al. (2020) models the effects of behavioural recommendations through reducing the number of daily effective contacts with respect to pre-confinement contact rates defined by Prem et al. (2017).

We denote by $C_l$ the number of contacts that occurred at a location $l$ during pre-confinement. While operating under recommendation level 0, an agent visiting location $l$ will on average contact $C_l$ other agents. Post-confinement contact patterns surveyed in Quebec during summer 2020 (Brisson & et al., 2020) are used to scale down the number of pre-confinement interactions across different locations, the percentage reduction per location are denoted $\alpha_l$. Table 2 show the number of contacts and percentage reduction as a result of confinement.

| Location | $C_l$ | $\alpha_l$ % |
|---|---|---|
| Household | 2.7 | 0.30 |
| Workplace | 10 | 0.80 |
| School | 6 | 0.80 |
| Other | 3.1 | 0.50 |

Table 2: **Daily contacts per location type:** This table shows for each location $l$ the pre-confinement mean number of daily contacts $C_l$ and the reduction in number of contacts $\alpha_l$ based on data collected in the Region of Montréal.

There are 5 recommendation levels described by Table 3 with their corresponding expected number of contacts. When these recommendations are followed, they modulate the number of effective contacts that agents have while at locations. Recommendation level 3 scales down the contacts at a location by $\alpha_l$. Levels 1 and 2 are successively obtained by reducing the location dependent contacts by half from level 3, while level 4 is used to impose full quarantine i.e. no contacts. Level 0 is not used except to represent the behavior when an agent does not adhere to the recommendations modeled by a daily dropout parameter.

| Recommendation Level | Description | Effective contacts |
|---|---|---|
| Level 0 | Pre-pandemic behaviour | $C_l$ |
| Level 1 | Intermediate recommendations reducing contacts | $1/4 * (1 - \alpha_l) * C_l$ |
| Level 2 | Stronger recommendations reducing contacts | $1/2 * (1 - \alpha_l) * C_l$ |
| Level 3 | Post-confinement behaviour | $(1 - \alpha_l) * C_l$ |
| Level 4 | Quarantine/self-isolation | $0$ |

Table 3: **Recommendation level and corresponding effective contacts:** Recommendation levels and corresponding expected number of daily effective contacts for a location type $l$.

## 5.6 INFECTIVITY (TRANSMISSION) MODEL

Effective viral load (EVL) varies [0., 1.], and represents the 'severity' of viral load the agent is currently experiencing (which takes into account both the viral load and the interaction of the virus with the host's immune system), as shown in Figure 8. Each agent's disease progresses according to this curve, and the probability of transmission is also proportional to EVL.

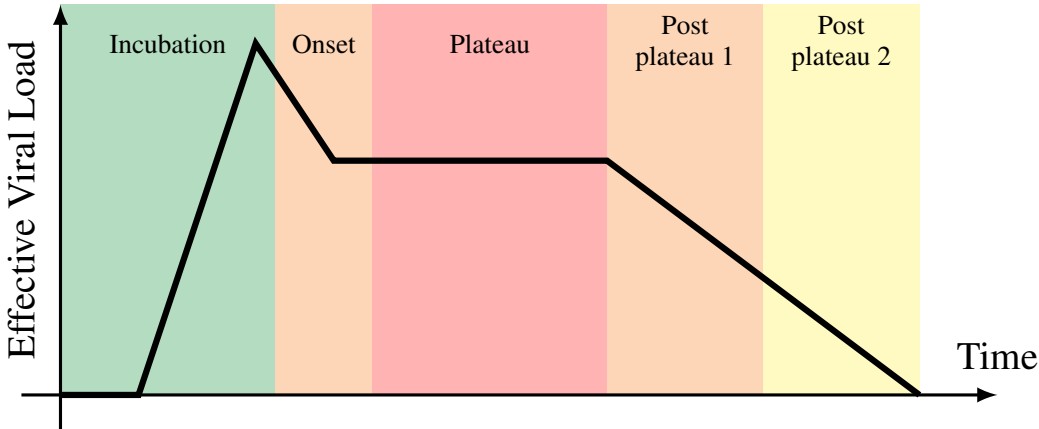

Figure 8: **Effective Viral Load** [Reproduced with permission from Gupta et al. (2020)]: Schematic showing the viral load curve, and associated phases of symptoms with severity indicators: infectiousness onset occurs on average 2.5 days after exposure, viral load peaks 0.7 days before symptom onset, which occurs an average of 5 **incubation days** after exposure. Symptoms are most severe after viral load peak and symptom onset, when the virus has had time to infect many cells. Recovery takes on average 14 days from symptom onset.

## 5.7 TAILORING THE SIMULATOR TO REGIONAL DEMOGRAPHICS AND OTHER DISEASES

Many aspects of the simulator are customizable to a particular region's statistics, including individual agent features (pre-existing medical conditions, biological sex, etc.), contact patterns, and prevalence of other diseases (which affect the predictions of COVID-19 by introducing other reasons for an agent to experience symptoms. The simulator provides details of fit to real data for the region of Montreal, Canada, with the confounding symptom-producing conditions of flu, allergies, and colds. The process for tailoring the simulator to other regions is detailed in their work, but briefly, configuration files with the appropriate statistics need to be provided in `https://github.com/mila-iqia/COVI-AgentSim/blob/master/src/covid19sim/configs/`.

