# OpenReview forum: "Predicting Infectiousness for Proactive Contact Tracing"
_ICLR.cc/2021/Conference — ICLR 2021 Spotlight_

### Official Review · AnonReviewer3 · 2020-10-28
**A great tool for informing public policy!**

**Rating:** 7
**Confidence:** 4

**Review:**

Summary: In this paper the authors propose a novel method of contact tracing which they dub Proactive Contact Tracing (PCT). PCT is aimed at detecting an individual's infectivity given locally observed information (and history), as can be carried on say a mobile device. As the name suggests, being able to estimate infectivity given this local information is useful in large part due to the fact that individuals can be asymptomatic, or they can have significant infectivity before demonstrating symptoms if they are to be symptomatic. Proactively estimating infectivity levels thus provides another tool in combatting viral spread through proper containment protocols.

The core methodology behind PCT lies in training a deep neural network offline with viral simulation data to estimate this very infectivity given local information / history. The purpose of this offline training is to preserve privacy of users, as they would only need to locally run the computation of their infectivity estimates (rather than send information to a centralised server for example for training). Furthermore, these infectivity estimates in practice can be used for containment strategies. The authors propose truncating the infectivity values to 4 bits, and using this information by a centralised agent to recommend different distancing for agents.

Finally, the authors back their results with simulations where they show that under some (unspecified) containment regime, the use of PCT can provide real gains over current contact tracing methods in terms of minimising the spread of the virus while simultaneously minimising the amount of containment required of a population.


Reasons for score:

I think this work is well motivated and of great potential use to policymakers in what will be a long fight against COVID. In addition, the care given to preserving privacy (in terms of the messages sent from devices to centralised agents) is of utmost importance, and these very constraints are necessary in the solution space the authors are navigating.


General questions / Points to address:

-I would like the authors to give more clarity on what the nature of the containment protocol they propose is. They mention that as a function of $y_i$, a centralised agent will propose $\zeta_i$, a containment measure for an agent, but I am curious what this entails. Is it the case that an agent is isolated for a certain amount of time? Do they reduce their number of contacts by a proportional amount? The reason I ask this is that contact tracing measures are only as useful as the containment protocol that they are paired with.
-I am also still a bit unclear as to what the infectivity is modelling specifically. The authors mention that $y_i$ is a positive real number, but what does this equate to in the underlying models used to train the DNN? I'm guessing this is an intrinsic parameter to the model generating data, but it would be useful to perhaps get an overview of the model itself if there is space.
-Have the authors considered how the upcoming flu season might affect PCA, especially if there are common symptoms between diseases? In other developing countries, such as central Mexico, Dengue season is also just beginning, and the co-infection of diseases is an issue that local authorities are grappling with for their contact tracing applications
-I also wonder if the authors have considered using the heterogeneity of the population to their advantage in terms of the underlying proposed containment protocol. Ultimately PCT is studying an individuals exposure data (albeit locally), and this very data can help understand the effictiveness of a proposed containment policy given their infectivity (such as an individuals propensity to listen to proposed containment, or whether this individual has a high baseline exposure to other individuals, and hence a possibility of spreading to many if they are infected). Recent work has focused on using heterogeneous population information to optimise pandemic policy (https://bfi.uchicago.edu/working-paper/socioeconomic-network-heterogeneity-and-pandemic-policy-response/), which could be of interest to the authors.

---

### Official Review · AnonReviewer1 · 2020-10-29
**sounds promising, hard to evaluate some aspects of the contribution**

**Rating:** 7
**Confidence:** 2

**Review:**

In the manuscript entitled "Predicting infectiousness from proactive contact tracing" the authors present a sophisticated algorithmic approach to control of app-based quarantine advisories with regard to covid-19.  The methodology described covers multiple aspects of this practical problem: privacy preservation for the mobile phone data, trade-offs between reducing transmission & limiting constraints on personal freedom, synthesis of multiple data sources (e.g. test results, self-reported symptoms etc).  As an experienced epidemiological modeller I can say that the description of each component of this manuscript read as sensible to me and that no 'red flags' were raised.  However, I cannot comment on the details of the privacy preservation methods and I was left unsure of the necessity of a deep learning model for this problem over a simpler (and hence more readily interpretable for public health policy makers) parametric statistical approach.

---

### Official Review · AnonReviewer2 · 2020-10-30
**An outstanding contribution to digital contact tracing**

**Rating:** 9
**Confidence:** 3

**Review:**

This paper introduces a deep learning based digital contact tracing method to minimize the spread of COVID19. The proposed method is based on locally processed information collected on the mobile app. Unlike the most commonly used digital tracing approach that sends quarantine recommendations to all recent contacts of a newly diagnosed person, the developed method in this paper considers all the information related to the users and the ones who have been in contacts with them in order to make user specific recommendations. This is not an easy problem because of different conflicting factors involving in making recommendation decisions, i.e. user privacy, mobility restrictions, and public health. The proposed method, called proactive content tracing, is a set-based architecture (that uses attention) and perform distributed inference to preserve privacy.

The authors utilized a simulator called COVIsim (not sure whether it is COVAsim or COVIsim) to train the proposed deep learning model and showed that their proposed method outperforms two baselines on multiple different metrics. The use of simulated data is well-justified given how difficult it is to obtain relevant COVID19 tracing data. This paper advances the current digital contact tracing significantly and is a great contribution to the field.

---

### Author Response · Authors · 2020-11-22
**Summary of changes in revision:**

1. Updated name of the simulator we use from COVIsim to COVI-AgentSim (as per their work)
2. Increased size of figures for improved readability
3. Mentioned in the conclusion the possibility of future work learning policies targetted to heterogeneous populations, as suggested by R3
4. Added Appendix 5 with relevant background information on the simulator, including:
    * More detail on the recommendation levels which determine the containment protocol modeled by the simulator
    * Summary of infectivity (transmission) model, and
    * How the demographics and other regional information for the simulator can be tailored to different regions

---

### Decision · Program_Chairs · 2021-01-07
**Final Decision**

**Decision:**

Accept (Spotlight)

**Comment:**

The reviewers unanimously agree that the paper is timely, well motivated and correct, with potential to significantly impact digital contact tracing.